# Assessing the Impact of Primary-Series Infection and Booster Vaccination on Protection against Omicron in Hong Kong: A Population-Based Observational Study

**DOI:** 10.3390/vaccines12091014

**Published:** 2024-09-05

**Authors:** Jialiang Jiang, Eric Ho Yin Lau, Ziyi Zhou, Guosheng Yin, Yun Lin, Benjamin John Cowling, Kwok Fai Lam

**Affiliations:** 1Department of Statistics and Actuarial Science, The University of Hong Kong, Pokfulam, Hong Kong SAR, China; u3558597@connect.hku.hk (J.J.); u3577083@connect.hku.hk (Z.Z.); gyin@hku.hk (G.Y.); 2Laboratory of Data Discovery for Health (D24H), Hong Kong Science and Technology Park, New Territories, Hong Kong SAR, Chinavanialam@connect.hku.hk (Y.L.); bcowling@hku.hk (B.J.C.); 3WHO Collaborating Centre for Infectious Disease Epidemiology and Control, School of Public Health, LKS Faculty of Medicine, The University of Hong Kong, Pokfulam, Hong Kong SAR, China; 4Centre for Quantitative Medicine, Duke-NUS Medical School, Singapore 169857, Singapore

**Keywords:** Comirnaty, CoronaVac, Omicron, SARS-CoV-2

## Abstract

This study aimed to assess the real-world effectiveness of vaccines and hybrid immunity in preventing infections during the Omicron prevalent period in Hong Kong. This study analyzed vaccination records and COVID-19 confirmed case records from 1 January 2022 to 28 January 2023 and included a total of 7,165,862 individuals with vaccination or infection records. This study found that an additional vaccine dose offered increased protection against Omicron BA.1/2 and BA.4 infections for individuals without prior infections in general. Hybrid immunity, acquired through vaccination and natural infection, was found to be significantly stronger than that provided by vaccines alone. The Comirnaty Original/Omicron BA.4/5 bivalent vaccine, introduced in December 2022, was associated with a lower risk of BA.4 infection when administered as a booster dose after three doses of CoronaVac. However, individuals with four doses of the CoronaVac vaccine did not exhibit a significantly lower risk of infection compared to those with three doses during the BA.4 dominant period. This study highlights the importance of promoting booster shot uptake and encouraging vaccination among those who have recovered from COVID-19 infections. The potential immune imprinting effect associated with the Comirnaty and CoronaVac vaccine underscores the need for continued surveillance and research to optimize vaccination strategies for emerging variants.

## 1. Introduction

The COVID-19 pandemic has posed unparalleled challenges to global healthcare systems, leading to millions of infections and fatalities across the globe. In Hong Kong, the pandemic had given rise to an urgent requirement for effective treatments to mitigate the spread of infections. Vaccines have emerged as vital interventions in reducing the impact of SARS-CoV-2. As time progresses, a growing number of individuals contract the virus which in turn provides some degree of immunity against infections/reinfections. This study aims to assess the real-world effectiveness of vaccines and hybrid immunity in preventing infections during the Omicron prevalent period in Hong Kong.

The COVID-19 pandemic, caused by the severe acute respiratory syndrome coronavirus 2 (SARS-CoV-2), has posed unprecedented challenges to global healthcare systems. It has led to an overwhelming number of reported infections and a staggering loss of lives worldwide. As of 20 February 2024, there have been in excess of 774 million reported infections and an estimated 7 million recorded fatalities. However, it is crucial to acknowledge that these reported figures are likely to be underestimated and the true extent of the impact may in fact be even greater [1,2,3]. The COVID-19 pandemic has generated an urgent requirement for effective therapeutics to mitigate the spread of infections, not only in Hong Kong but also in other parts of the world. The global scientific community has been deeply engaged in an unparalleled endeavor to address the impact of the SARS-CoV-2 virus. These collective challenges have led to the rapid development of vaccines and the repurposing of existing medications for combating the disease. Among these interventions, vaccines have emerged as critical tools in reducing the occurrence of severe cases, lowering infection rates, mitigating disease progression, and decreasing mortality rates among hospitalized patients [4].

Since 23 February 2021, the Hong Kong government has implemented a territory-wide COVID-19 Vaccination Program, aiming to increase the vaccination rate among all residents [5,6,7,8]. This program aims to provide a scientifically based vaccination arrangement for the population, with the goal of safeguarding public health and restoring normal societal activities. Two types of vaccines have been authorized for use in Hong Kong: Comirnaty (BNT162b2), an mRNA-based vaccine, and CoronaVac (Sinovac), an inactivated virus vaccine. Both vaccines have shown good protective efficacy against symptomatic SARS-CoV-2 infection and severe COVID-19 [9,10,11,12,13,14].

While vaccines have played a crucial role in combating COVID-19 by lowering the risks of infections, complications, and deaths, the SARS-CoV-2 virus has displayed rapid mutation, evading immune responses generated by previous infections and vaccines [15,16,17]. This has resulted in increased transmission rates compared to earlier variants. The World Health Organization (WHO) identified certain variants, namely Alpha, Beta, Gamma, Delta, and Omicron, as variants of concern (VOCs) due to their heightened threat to global public health. As the virus continues to evolve and the effectiveness of vaccines diminishes against circulating variants, Hong Kong has experienced six waves of the COVID-19 pandemic [18,19]. There were only 12,631 confirmed cases and 213 fatal cases in the first four waves of the pandemic, which is not as serious compared to the later waves [20]. The fifth and sixth waves occurred between 31 December 2021, and 1 January 2023, resulting in a reported 2,863,475 infection cases and 12,921 deaths. Throughout the pandemic, the SARS-CoV-2 virus has undergone mutations, giving rise to numerous variants. Each wave in Hong Kong was predominantly driven by a specific lineage, except for the first and second waves. Genomic sequencing analysis revealed that the third and fourth waves in Hong Kong were dominated by B.1.1.63 and B.1.36.27 lineages, respectively, which had evolved from the original wild-type virus. The circulation of SARS-CoV-2 remained relatively low in Hong Kong until the emergence of the Omicron (B.1.1.529) sublineage BA.2.2 which triggered the fifth wave in January 2022. Subsequently, the BA.4/5 variant emerged in June 2022 and became the predominant strain in August 2022, leading to the sixth wave of the pandemic [21,22].

Given the complexity of the situation, designing a vaccination plan for individuals who have already been vaccinated or have previously been infected becomes crucial. As the SARS-CoV-2 virus evolved, it underwent rapid mutations, which required the administration of more vaccine doses to maintain adequate protective efficacy [23]. In a retrospective cohort study, Chemaitelly et al. found that individuals who have received one or more additional vaccine doses may have reduced protection against reinfection during the period when the Omicron variant was predominant, potentially due to immune imprinting [24]. This may be related to the IgG4 antibodies induced by repeated vaccination [25]. However, this pattern was not observed during their previous study when the Alpha, Beta, and Delta variants were predominant [26]. Monge et al. suggested that the increased risk of reinfection in vaccinated individuals with a booster compared to those without a booster may be due to selection bias rather than immune imprinting of the COVID-19 vaccines [27]. Hence, our objective is to employ a statistical methodology to explore the potential occurrence of immune imprinting in Hong Kong and identify the optimal vaccination strategy against the novel variant, thereby empowering us to learn from past experiences and effectively prepare for future incidents.

## 2. Materials and Methods

### 2.1. Study Design and Study Population

In this territory-wide study, we analyzed vaccination records obtained from the Hong Kong Department of Health. These records provided information on individuals, including their sex, vaccination types, and vaccination dates. We also collected data on COVID-19 confirmed case records from the Hong Kong Centre for Health Protection. These datasets were combined and analyzed using unique pseudo identifiers in order to gain comprehensive insights into the relationship between vaccination and infection/reinfection against SARS-CoV-2. Three vaccine products were considered: Comirnaty, CoronaVac, and the Comirnaty Original/Omicron BA.4/5 bivalent vaccine, referred to as B, S, and O, respectively. Additionally, infections with two types of Omicron strains were considered: BA.1/2 and BA.4, denoted as P and A, respectively. Regarding the explanatory variables, age was categorized into three groups: 18–59, 60–79, and 80 and above, with the 18–59 age group serving as the reference category. The reference level for sex was set as female.

Our dataset consisted of a total of 7,165,862 individuals with vaccination or infection records from 1 January 2022 to 28 January 2023. Among these, 6,296,183 were adults. The objective of this study was to compare the effectiveness of vaccines and hybrid immunity in preventing Omicron infection/reinfection, as well as to assess the efficacy of a new version of the vaccine compared to the old version. However, it is important to note that during the study period, there were multiple dominant subvariants of Omicron. Due to the mixed dominance of BA.1/2 and BA.4 between 1 July 2022 and 30 September 2022, and the limited availability of detailed infection data, we considered three different scenarios for different purposes [28].

Scenario I: The objective of Scenario I was to investigate whether immune imprinting occurred under the Omicron BA.1/2 subvariant. The study period spanned the dominant period of BA.1/2, from 1 January 2022 to 30 June 2022 [22]. To align with the vaccination guidelines provided by the Hong Kong government during that period, we examined two common vaccination scenarios: individuals who received two doses, and those who received three doses [29]. Given our interest in the combined effect of vaccination followed by Omicron infection, the analysis considered a total of eight vaccination–infection profiles, accounting for the sequence of vaccination and infection: BB, BBB, BBP, BBBP, SS, SSS, SSP, and SSSP. (e.g., SSP profile describes an individual who took two doses of CoronaVac, and then later contracted a BA.1/2 infection). To facilitate the analysis, a random sample of 1,500,000 adults was drawn. Within this sample, 785,940 individuals were in the Comirnaty vaccination profiles (BB, BBB, BBP, BBBP), while 562,569 were in the CoronaVac vaccination profiles (SS, SSS, SSP, SSSP).

Scenario II: The objective of Scenario II was to investigate whether immune imprinting occurred under the new Omicron BA.4 subvariant. The study period spanned the dominant period of BA.4, from 31 October 2022 to 28 January 2023 [22]. To align with the vaccination guidelines provided by the Hong Kong government during that period, we examined three common vaccination scenarios: individuals who received two, three or four doses. Considering the combined effect of two or three doses of vaccines in combination with Omicron BA.1/2 or BA.4 infections, and accounting for the two vaccine products, the analysis included a total of twelve vaccination–infection profiles: BBB, BBBB, BBP, BBBP, BBA, BBBA, SSS, SSSS, SSP, SSSP, SSA, and SSSA. A total of 2,328,238 individuals were in the Comirnaty three-dose and four-dose vaccination profiles (BBB, BBBB), while 1,492,833 were in the CoronaVac three-dose and four-dose profiles (SSS, SSSS). Additionally, 350,502 individuals were in the BBP and BBBP profiles, 298,535 were in the BBA and BBBA profiles, 201,572 were in the SSP and SSSP profiles, and 155,907 were in the SSA and SSSA profiles.

Scenario III: The objective of Scenario III was to investigate the relative effectiveness of booster doses using the Comirnaty bivalent vaccine compared to the original vaccine type under the new Omicron BA.4 subvariant. It is noteworthy that the Comirnaty Original/Omicron BA.4/5 bivalent vaccine was introduced in Hong Kong on 1 December 2022 [30]. The study period spanned the dominant period of the BA.4 subvariant, from 1 December 2022 to 28 January 2023 [22]. We examined two common vaccination scenarios: individuals who received three doses and those who received four doses. The vaccination profiles considered were BBB, BBBB, BBBO, SSS, SSSS, and SSSO, accounting for the sequence of vaccination. A total of 2,067,139 individuals were in the three-dose BBB profile, 187,365 were in the four-dose BBBB profile, and 103,280 were in the BBBO profile. Furthermore, 1,232,124 individuals were in the three-dose SSS profile, 248,830 were in the four-dose SSSS profile, and 15,978 were in the SSSO profile.

The study periods for the three scenarios are illustrated in Figure 1. It is worth noting that we assumed that individuals would not be infected within one month since last vaccination or infection. Therefore, the follow-up period commenced one month after individuals received vaccination or became infected, and it continued until the occurrence of the primary outcome of interest (i.e., infection), the administration of an additional vaccination, or the previously defined observation period as in Scenarios I, II, and III.

### 2.2. Statistical Analysis

To mitigate the potential issue of immortal time bias and account for time-varying infection risk and vaccination status, we organized and analyzed the risk set according to calendar days. Due to the presence of recurrent events, the effects of time-constant vaccine and primary infection were analyzed by employing the Andersen–Gill model, an extension of the proportional hazards model proposed by Cox [31,32,33,34]. The hazard of infection on day
t is provided by
(1)$$\lambda \left(t\right)=\lambda_{0}\left(t\right)\exp\left[\gamma^{T}x\left(t\right)\right],$$ where
$\lambda_{0}\left(t\right)\geq 0$ is the baseline hazard function that characterizes the varying infection rates over different phases of the pandemic;
$x\left(t\right)$ is a vector of covariates including age, sex, and vaccination-infection status; and
γ represents the corresponding coefficients for these covariates. The partial likelihood using Breslow’s method for ties is adopted for the estimation of
γ.

## 3. Results

### 3.1. Scenario I

A total of 1,348,509 individuals were identified from the randomly drawn sample who had received two or three doses of a single-type vaccine, with or without experiencing any subsequent Omicron BA.1/2 infection at the time of enrollment. Figure A1 depicts the daily number of infections for the Comirnaty and CoronaVac cohorts from the sampled data. The corresponding modeling results for the Comirnaty and CoronaVac cohorts are presented in Table 1 and Table 2.

Comparing individuals with different vaccination–infection profiles, including BB, BBB, BBP, and BBBP, it is found that the group with the BB profile exhibits a higher risk of Omicron BA.1/2 infection. The hazard ratios for these groups, compared to the BB group, are 0.513, 0.030, and 0.057, respectively. These findings indicate that an additional dose of the vaccine provides increased protection against Omicron BA.1/2 infection for individuals who have not been previously infected. Furthermore, in terms of protection, immunity acquired from infection with the BA.1/2 variant confers a significantly stronger level of defense compared to an additional dose of vaccines. However, when comparing individuals with different vaccination–infection profiles (BB, BBB, BBP, and BBBP), the study reveals that the BBBP profile carries a higher risk of Omicron BA.1/2 reinfection compared to the BBP profile, with a hazard ratio of 1.886 (95% CI 1.056 to 3.366, *p* = 0.003).

When comparing individuals with different vaccination–infection profiles, namely, SS, SSS, SSP, and SSSP, it is found that those in the SS group have a significantly higher risk of Omicron BA.1/2 infection. The hazard ratios for these groups, compared to the SS group, are 0.707, 0.048, and 0.067, respectively. These results indicate that an additional dose of the CoronaVac vaccine provides increased protection against Omicron BA.1/2 infection for individuals who have not been previously infected. Furthermore, compared to the protection provided by vaccines, the immunity gained from a primary BA.1/2 infection offers a significantly stronger level of defense. However, when comparing individuals with different vaccination–infection profiles, specifically those in the SSP group to those in the SSSP group, it is observed that the latter group has a higher risk of Omicron BA.1/2 reinfection, but the difference is not statistically significant (hazard ratio for SSSP vs. SSP: 1.381, 95% CI 0.796 to 2.394, *p* = 0.251). This observation aligns with previous studies conducted in Qatar, indicating a potential immune imprinting effect for Omicron BA.1/2 reinfection [21].

### 3.2. Scenario II

A total of 3,821,071 individuals had received three or four doses of a single-type vaccine without subsequent infection by the Omicron BA.1/2 or BA.4 variants at the time of enrollment, while 1,006,516 individuals had received two or three doses of a single-type vaccine with subsequent infection by the Omicron BA.1/2 or BA.4 variants. Figure A2 illustrates the daily number of infections for the Comirnaty and CoronaVac cohorts. The corresponding modeling results for the Comirnaty and CoronaVac cohorts are presented in Table 3 and Table 4.

Comparing individuals with different vaccination–infection profiles, including BBB, BBBB, BBP, BBBP, BBA, and BBBA, those in the BBB group had a significantly higher risk of Omicron BA.4 infection. The hazard ratios for these groups, compared to the BBB group, were 0.892, 0.266, 0.260, 0.007, and 0.004, respectively. These results indicate that receiving an additional dose of the Comirnaty vaccine provides higher protection efficacy against Omicron BA.4 infection compared to three doses for individuals who have not been previously infected. However, the immunity gained from either the BA.1/2 or BA.4 virus offers much stronger protection efficacy compared to the vaccine.

On the other hand, when comparing individuals with different vaccination–infection profiles, including BBP and BBBP, the latter group had a lower risk of Omicron BA.4 infection, but the difference was not statistically significant (hazard ratio for BBBP vs. BBP: 0.975, 95% CI 0.943 to 1.008, *p* = 0.138). Similarly, when comparing individuals with different vaccination–infection profiles, including BBA and BBBA, the latter group had a lower risk of Omicron BA.4 reinfection, but the difference was also not statistically significant (hazard ratio for BBBA vs. BBA: 0.613, 95% CI 0.248 to 1.514, *p* = 0.289).

In contrast, individuals in the SSSS group exhibited a significantly higher risk of Omicron BA.4 infection compared to those in the other vaccination–infection profiles, including SSS, SSP, SSSP, SSA, and SSSA. The hazard ratios for these groups, compared to the SSS group, were as follows: 1.226, 0.357, 0.312, 0.008, and 0.005, respectively. These findings suggest that receiving an additional dose of the CoronaVac vaccine offers lower protection against Omicron BA.4 infection compared to individuals who have not been previously infected and have received three doses. This observation is consistent with studies conducted in China and indicates that repeated vaccination with an inactivated vaccine may diminish the effectiveness of neutralizing antibodies against Omicron variants in breakthrough infections [34].

When comparing the protection from the vaccine to the immunity gained from either the BA.1/2 or BA.4 virus, the immunity from the virus provides much stronger protection efficacy. On the other hand, when comparing individuals with different vaccination–infection profiles, including SSP and SSSP, the latter group had a significantly lower risk of Omicron BA.4 infection (hazard ratio for SSSP vs. SSP: 0.873, 95% CI 0.837 to 0.910, *p* < 0.001). Similarly, when comparing individuals with different vaccination–infection profiles, including SSA and SSSA, the latter group had a lower risk of Omicron BA.4 reinfection, but the difference was not statistically significant (hazard ratio for SSSA vs. SSA: 0.614, 95% CI 0.220 to 1.714, *p* = 0.352). This observation indicates that no immune imprinting effect was observed during the BA.4 dominant period.

### 3.3. Scenario III

The study population comprised 2,254,504 individuals who received three or four doses of Comirnaty at the time of enrollment and 1,480,954 individuals received three or four doses of CoronaVac. Furthermore, 203,343 individuals received three doses of a single-type vaccine followed by one dose of the Comirnaty bivalent vaccine. For clarity, we refer to individuals who received the first three doses of the Comirnaty vaccine as the Comirnaty cohort, and individuals who received the first three doses of the CoronaVac vaccine as the CoronaVac cohort. The daily numbers of infections for the two cohorts are illustrated in Figure A3. The corresponding modeling results for Comirnaty and CoronaVac cohorts are presented in Table 5 and Table 6.

Comparing individuals with different vaccination–infection profiles, including BBB, BBBB, and BBBO, those in the BBB group had a significantly higher risk of Omicron BA.4 infection. The respective hazard ratios for these groups, compared to the BBB group, were 0.881 and 0.757, respectively. When comparing individuals with different vaccination–infection profiles, including BBBB and BBBO, the latter group had a significantly lower risk of Omicron BA.4 infection (hazard ratio for BBBO vs. BBBB: 0.859, 95% CI 0.814 to 0.907, *p* < 0.001). This finding suggests that the bivalent vaccine is more effective than the monovalent vaccine against the BA.4 strain.

Similarly, when comparing individuals with different vaccination–infection profiles, including SSS, SSSS, and SSSO, those in the SSSS group had a higher risk of Omicron BA.4 infection. The respective hazard ratios for these groups, compared to the SSS group, were 1.223 and 0.796, respectively. Similar to scenario II, it was observed that repeated administration of the CoronaVac vaccine may potentially diminish the efficacy of protection against the Omicron BA.4 variant. When comparing individuals with different vaccination–infection profiles, including SSSS and SSSO, the latter group had a significantly lower risk of Omicron BA.4 infection (hazard ratio for SSSO: 0.651, 95% CI 0.555 to 0.763, *p* < 0.001). This discovery suggests that employing the more recent Comirnaty bivalent vaccine could be a viable strategy to enhance the protection efficacy for individuals who have received the CoronaVac vaccine.

## 4. Discussion

This territory-wide observational study provides valuable insights into the real-world effectiveness of COVID-19 vaccines and hybrid immunity against Omicron infections in Hong Kong. The analysis based on a large population dataset, encompassing over 7 million individuals with vaccination or infection records, offered a comprehensive understanding of the relationship between vaccination, natural infection, and protection against SARS-CoV-2.

A notable finding from our study is the increased protection conferred by an additional vaccine dose against Omicron BA.1/2 and BA.4 infections for individuals without prior infections. However, during the BA.4 dominant period, individuals with four doses of the CoronaVac vaccine did not exhibit a significantly lower risk of infection compared to those with three doses. Interestingly, the Comirnaty Original/Omicron BA.4-5 bivalent vaccine, introduced in December 2022, was associated with a lower risk of BA.4 infection when administered as a booster dose after three doses of CoronaVac. These findings align with previous research demonstrating the benefits of booster doses in enhancing immune responses and reducing infection rates [35]. Notably, the protection offered by hybrid immunity, acquired through both vaccination and natural infection, was found to be significantly stronger than that provided by vaccines alone, consistent with observations in other regions [36].

Our results suggest a potential immune imprinting effect associated with the Comirnaty and CoronaVac vaccine during the BA.1/2 dominant period. Individuals who received three doses of Comirnaty or CoronaVac followed by a BA.1/2 infection had a higher risk of a BA.1/2 reinfection compared to those with two doses and a BA.1/2 infection. However, this pattern was not observed during the BA.4 dominant period, possibly due to the different subvariants being compared. The immune imprinting effect aligns with the findings of Chemaitelly et al., who reported reduced protection against reinfection during the Omicron period in individuals who received two or three vaccine doses [24]. Nonetheless, there may be alternative explanations for the observed differences between the vaccination groups [27].

While our study provides valuable insights, certain limitations should be acknowledged. First, our analysis did not account for potential confounding factors, such as underlying health conditions or socioeconomic status, which may influence the risk of infection and disease severity. Second, the study period spanned multiple Omicron subvariants, and the effectiveness of vaccines and hybrid immunity may vary across these different subvariants. Third, this study focuses only on reported COVID-19 cases, and any unreported infections may have influenced the results. Additionally, our study did not consider the potential impact of waning immunity over time, which could affect the observed levels of protection.

Despite these limitations, our findings have important implications for public health policies and vaccination strategies. The observed benefits of additional vaccine doses, particularly the bivalent Comirnaty vaccine, and hybrid immunity highlight the importance of promoting booster shot uptake. Furthermore, the potential immune imprinting effect associated with the Comirnaty vaccine underscores the imperative need for ongoing surveillance and rigorous research aimed at optimizing vaccination strategies in the face of emerging variants.

As the COVID-19 pandemic continues to evolve, it is crucial to remain vigilant and adjust public health interventions based on emerging scientific evidence. Data from the Japanese health ministry revealed that the average number of new COVID-19 infections at designated hospitals across the country rose for 10 consecutive weeks in the weeks through 14 July 2024. Furthermore, the chairperson of the Japanese Association for Infectious Diseases reported that Japan is likely entering an 11th wave of COVID-19 infections dominated by the KP.3 variant, which was derived from the Omicron JN.1 strain since early 2024. The KP.3 variant is considered more contagious and capable of evading immunity acquired through previous infections and vaccinations [37]. Future research should focus on elucidating the mechanisms underlying immune imprinting and exploring strategies to mitigate its potential impact on vaccine effectiveness, such as the development of variant-specific vaccines. Additionally, ongoing monitoring of vaccine performance against emerging variants is essential to inform timely updates to vaccination policies and recommendations. As the pandemic continues to evolve, this evidence-based approach will be crucial in guiding public health decision-making and protecting populations from the impact of COVID-19.

## 5. Conclusions

Our study provides valuable insights into the real-world effectiveness of COVID-19 vaccines and hybrid immunity against Omicron infections in Hong Kong. While additional vaccine doses, particularly the bivalent Comirnaty vaccine, and hybrid immunity offer increased protection, the potential immune imprinting effect observed with the Comirnaty vaccine warrants further investigation to optimize vaccination strategies and maintain robust immune responses against emerging SARS-CoV-2 variants.

## Figures and Tables

**Figure 1 vaccines-12-01014-f001:**
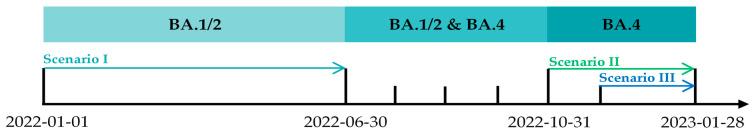
The study period for Scenarios I, II, and III.

**Table 1 vaccines-12-01014-t001:** Estimation results for Scenario I (Comirnaty cohort).

Variable	Against Omicron BA.1/2 Infection		
	Hazard Ratio ^a^	95% CI	*p* Value
Age 18–59 (Reference)	-	-	-
Age 60–79	0.932	(0.915, 0.950)	<0.001
Age 80 and above	0.691	(0.642, 0.744)	<0.001
Female (Reference)	-	-	-
Male	1.132	(1.117, 1.147)	<0.001
BB ^b^ (Reference)	-	-	-
BBB	0.513	(0.503, 0.523)	<0.001
BBP ^c^	0.030	(0.022, 0.042)	<0.001
BBBP	0.057	(0.035, 0.092)	<0.001

^a^ Hazard ratio is calculated by comparing the current level of the variable with the corresponding reference level. ^b^ B represents the Comirnaty vaccine. ^c^ P represents the BA.1/2 infection.

**Table 2 vaccines-12-01014-t002:** Estimation results for Scenario I (CoronaVac cohort).

Variable	Against Omicron BA.1/2 Infection		
	Hazard Ratio ^a^	95% CI	*p* Value
Age 18–59 (Reference)	-	-	-
Age 60–79	0.976	(0.959, 0.993)	0.005
Age 80 and above	0.845	(0.810, 0.882)	<0.001
Female (Reference)	-	-	-
Male	1.088	(1.071, 1.106)	<0.001
SS ^b^ (Reference)	-	-	-
SSS	0.707	(0.695, 0.721)	<0.001
SSP ^c^	0.048	(0.034, 0.069)	<0.001
SSSP	0.067	(0.044, 0.103)	<0.001

^a^ Hazard ratio is calculated by comparing the current level of the variable with the corresponding reference level. ^b^ S represents the CoronaVac vaccine. ^c^ P represents the BA.1/2 infection.

**Table 3 vaccines-12-01014-t003:** Estimation results for Scenario II (Comirnaty cohort).

Variable	Against Omicron BA.4 Infection		
	Hazard Ratio ^a^	95% CI	*p* Value
Age 18–59 (Reference)	-	-	-
Age 60–79	0.818	(0.810, 0.825)	<0.001
Age 80 and above	0.783	(0.762, 0.806)	<0.001
Female (Reference)	-	-	-
Male	0.999	(0.992, 1.005)	0.656
BBB ^b^ (Reference)	-	-	-
BBBB	0.892	(0.880, 0.904)	<0.001
BBP ^c^	0.266	(0.261, 0.272)	<0.001
BBBP	0.260	(0.253, 0.267)	<0.001
BBA ^d^	0.007	(0.003, 0.016)	<0.001
BBBA	0.004	(0.003, 0.005)	<0.001

^a^ Hazard ratio is calculated by comparing the current level of the variable with the corresponding reference level. ^b^ B represents the Comirnaty vaccine. ^c^ P represents the BA.1/2 infection. ^d^ A represents the BA.4 infection.

**Table 4 vaccines-12-01014-t004:** Estimation results for Scenario II (CoronaVac cohort).

Variable	Against Omicron BA.4 Infection		
	Hazard Ratio ^a^	95% CI	*p* Value
Age 18–59 (Reference)	-	-	-
Age 60–79	0.874	(0.866, 0.882)	<0.001
Age 80 and above	1.106	(1.090, 1.123)	<0.001
Female (Reference)	-	-	-
Male	0.940	(0.931, 0.949)	<0.001
SSS ^b^ (Reference)	-	-	-
SSSS	1.226	(1.212, 1.241)	<0.001
SSP ^c^	0.357	(0.346, 0.369)	<0.001
SSSP	0.312	(0.304, 0.321)	<0.001
SSA ^d^	0.008	(0.003, 0.022)	<0.001
SSSA	0.005	(0.004, 0.007)	<0.001

^a^ Hazard ratio is calculated by comparing the current level of the variable with the corresponding reference level. ^b^ S represents the CoronaVac vaccine. ^c^ P represents the BA.1/2 infection. ^d^ A represents the BA.4 infection.

**Table 5 vaccines-12-01014-t005:** Estimation results for Scenario III (Comirnaty cohort).

Variable	Against Omicron BA.4 Infection		
	Hazard Ratio ^a^	95% CI	*p* Value
Age 18–59 (Reference)	-	-	-
Age 60–79	0.808	(0.800, 0.817)	<0.001
Age 80 and above	0.793	(0.768, 0.819)	<0.001
Female (Reference)	-	-	-
Male	1.007	(1.000, 1.015)	0.056
BBB ^b^ (Reference)	-	-	-
BBBB	0.881	(0.868, 0.895)	<0.001
BBBO ^c^	0.757	(0.719, 0.798)	<0.001

^a^ Hazard ratio is calculated by comparing the current level of the variable with the corresponding reference level. ^b^ B represents the Comirnaty vaccine. ^c^ O represents the Comirnaty Original/Omicron BA.4/5 bivalent vaccine.

**Table 6 vaccines-12-01014-t006:** Estimation results for Scenario III (CoronaVac cohort).

Variable	Against Omicron BA.4 Infection		
	Hazard Ratio ^a^	95% CI	*p* Value
Age 18–59 (Reference)	-	-	-
Age 60–79	0.867	(0.857, 0.876)	<0.001
Age 80 and above	1.158	(1.139, 1.177)	<0.001
Female (Reference)	-	-	-
Male	0.939	(0.930, 0.948)	<0.001
SSS ^b^ (Reference)	-	-	-
SSSS	1.223	(1.207, 1.239)	<0.001
SSSO ^c^	0.796	(0.679, 0.933)	0.005

^a^ Hazard ratio is calculated by comparing the current level of the variable with the corresponding reference level. ^b^ S represents the CoronaVac vaccine. ^c^ O represents the Comirnaty Original/Omicron BA.4/5 bivalent vaccine.

## Data Availability

The data custodians (the Hospital Authority and the Department of Health of the Government of the Hong Kong Special Administrative Region) provided the underlying individual-patient data to the University of Hong Kong for the purpose of scientific research for this study. Restrictions apply to the availability of these data, which were used under license for this study. Authors must not transmit or release the data, in whole or in part, and in whatever form or media, or to any other parties or place outside of Hong Kong and must fully comply with the duties under the law relating to the protection of personal data, including those under the Personal Data (Privacy) Ordinance and its principles in all aspects.

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
