# Peer review of "Assessing the Impact of Primary-Series Infection and Booster Vaccination on Protection against Omicron in Hong Kong: A Population-Based Observational Study"

_vaccines, 2024, doi:10.3390/vaccines12091014_

Round 1

Reviewer 1 Report

Comments and Suggestions for Authors

The authors are trying to assess the real-world effectiveness of vaccines and hybrid immunitywith the use of vaccination records and COVID-19 confirmed case records in Hong Kong from January 1, 2022, to January 28, 2023, 18. Nevertheless, the efficiency of vaccinations cannot be estimated without taking into account the period before January 1, 2022, when the percentages of fully vaccinated people V and boosters B were much lower. As of January 1, 2022, the accumulated numbers of cases and deaths were C1=12667 and  D1=213, respectively at vaccination levels V1=62.3% and B1=5.28%, [*]. In 2022 the numbers of cases and deaths increased drastically despite much higher levels of vaccinations. In particular, as of December 4, 2022 the corresponding figures were: C2=2157976; D2=1086; V2=90.43%; B2=84.84%, [*]. It means that during 11 months of 2022 the registered numbers of cases were 169 times higher ((2157976-12667)/12667) and the number of deaths were 4 times higher ((1086-213)/213) than during previous 23 months of the COVID-19 pandemic. Thus, all vaccines have not decreased the numbers of cases and COVID-19 related deaths in Hong Kong. Similar situation is typical for other countries [**, ***]. Existing vaccines can reduce only the case fatality risks CFR (the deaths to case ratio). In particular, CFR1 = 213/12667= 0.0168 and CFR2 =(1086-213)/(2157976-12667)=  4.0693e-004. It means that in Hong Kong the probability to die for a person tested positive decreased  41 times in 2022 (in comparison the previous period of pandemic) due to vaccinations and natural immunity. Similar tends are visible for other countries [**, ***].

The study, which takes into account only the number of cases, cannot highlight “the importance of promoting booster shot uptake and encouraging vaccination among those who have recovered from COVID-19 infections”.  The CFR analysis is necessary “to optimize vaccination strategies for emerging variants”. The model used in the paper (eq. (1)) is unclear. In particular, the meaning of parameter T has be presented.  

[*] COVID-19 Data Repository by the Center for Systems Science and Engineering (CSSE) at Johns Hopkins University (JHU).  https://github.com/owid/covid-19-data/tree/master/public/data

  [**] Nesteruk I. (2024) Trends of the COVID-19 dynamics in 2022 and 2023 vs. the population age, testing and vaccination levels. Front. Big Data 6:1355080. doi: 10.3389/fdata.2023.1355080   [***]Nesteruk I. Should we ignore SARS-CoV-2 disease? Epidemiology and Infection. 2024;152:e57. doi:10.1017/S0950268824000487

Reviewer 2 Report

Comments and Suggestions for Authors

Current SARS-CoV-2 vaccines have limited durability and efficacy; this is partly due to viral variants.  In March, 2020 a preprint (later published) predicted that vaccines targeting the Spike protein would have limit durability due to variants.  Appropriate literature on this has not been cited or discussed.

IgG4 class switching from repeated COVID-19 vaccinations and booster shots is not presented.  Please update with inclusion of relevant citations.

This article neglects relevant adverse events associated with COVID-19 vaccinations.  A brief overview is warranted in the context of promotion of these vaccines.

Please expand the tables presented to include the number of patients and totals for relevant subgroups - this could take the form of one additional column in for format of N/M.

This study neglects the time dimension of when infections occur in the context of date of last vaccine or booster shot.  This is likely very important to the patterns of responses reported and associated observations made by the authors.

If additional booster shots will be needed for future strains, what is the strategy for avoiding immune imprinting, IgG4 antibody class switching, and antibody dependent enhancement (ADE)?

Round 2

Reviewer 1 Report

Comments and Suggestions for Authors

Unfortunately, the authors did not take into account my comments. They still believe
that “
the number of cases would be astronomically larger than those we observed”.
But 2.1 million cases registered cases during first 11 months of 2022
(with the high vaccination level) is really astronomically larger (169 times)
than during previous 23 months (with the low vaccination level) of the pandemic.
Since the population of Hong Kong  in  2022 was 7.3 million, we cannot state
that vaccinations have reduced the number of cases.
    I can agree only with the statement that “the number COVID-19-related deaths
would be astronomically larger than those we observed”. But the authors do not
consider the numbers of deaths and CFR.
This
makes the manuscript not only uninteresting, but also harmful to
understanding the effectiveness of vaccinations. I recommend to reject it.

Author Response

Comment: Unfortunately, the authors did not take into account my comments. They still believe that “the number of cases would be astronomically larger than those we observed”. But 2.1 million cases registered cases during first 11 months of 2022 (with the high vaccination level) is really astronomically larger (169 times) than during previous 23 months (with the low vaccination level) of the pandemic. Since the population of Hong Kong in 2022 was 7.3 million, we cannot state that vaccinations have reduced the number of cases. I can agree only with the statement that “the number COVID-19-related deaths would be astronomically larger than those we observed”. But the authors do not consider the numbers of deaths and CFR. This makes the manuscript not only uninteresting, but also harmful to understanding the effectiveness of vaccinations. I recommend to reject it.

Response:

Unfortunately, we cannot agree with statement that “all vaccines have not decreased the numbers of cases and COVID-19 related deaths in Hong Kong” in the first review.

The significant increase in the number of infections in 2022 was primarily due to the emergence of the mutated variant Omicron, during which the infection rate was 6 to 8 times higher than during the Delta predominant period. The effectiveness of vaccines is generally measured by protective efficacy, a relative measure compared to a certain reference group, such as the unvaccinated group.  For instance, the infection rate (measured under a certain unit, say 100,000 individuals per day) in the vaccinated group is 5 while in the unvaccinated group it is 10 during Delta predominant period, this indicates that the vaccine is effective in reducing the risk of infection. However, during the Omicron predominant period, the infection rate in the vaccinated and unvaccinated groups could be 500 and 1,000, respectively.  Thus, an increase in the total number of infections in the community does not imply that vaccines cannot reduce the numbers of cases and deaths. A more relevant question is “What if there were no vaccine?” This is a big question! If there were no vaccine, the pandemic might be still going strong worldwide. However, this is a counterfactual situation that cannot be addressed easily. We do not have a reference group (e.g., unvaccinated group) as in a randomized controlled trial.

We focus on time-to-event (here the event is infection) analysis using the extended version of the Cox proportional hazards model (known as the Andersen-Gill model). Based on our model, similar to many other research studies (e.g., see Lin et al. (2022a, 2022b, 2022c) below), we conclude that vaccinations provide reasonable protection against infection compared to the reference group even during the Omicron predominant period.

Finally, our primary focus is on the level of protection against infection, not against death. We do not understand why we should follow the two papers recommended by Reviewer 1 to study the case fatality rate (CFR). Reviewer 1 analyzed binary data only (i.e., data truncated at a particular time point), while our work focuses on the dynamic evolvement of the vaccine-infection process using survival analysis skills. It is well known that time-to-event analysis provides much more information than analysis using binary data. Different studies may have different objectives and use different statistical models. It is unfair that the paper is recommended for rejection because our findings are not in line with the reviewer’s previous findings based on entirely different objectives and methods of analyses.

Intellectual discussions are most welcome; however, drawing conclusions for the authors based on the cumulative infection and death numbers in Hong Kong is neither objective nor scientifically valid. In fact, we believe Reviewer 1’s conclusion “all vaccines have not decreased the numbers of cases and COVID-19 related deaths in Hong Kong” is unsound, because he/she ignored many factors that had caused the rising number of infections in Hong Kong or worldwide. A sound analysis should dig into the reasons and more advance statistical models, rather than relying upon simple ratios to draw conclusions.

References:

Lin DY, Gu Y, Wheeler B, Young H, Holloway S, Sunny SK, Moore Z, Zeng D. Effectiveness of Covid-19 vaccines over a 9-month period in North Carolina. New England Journal of Medicine 2022a; 386: 933–941.

Lin DY, Gu Y, Xu Y, Zeng D, Wheeler B, Young H, Sunny SK, Moore Z. Effects of vaccination and previous infection on omicron infections in children. New England Journal of Medicine 2022b; 387: 1141–1143.

Lin DY, Gu Y, Xu Y, Wheeler B, Young H, Sunny SK, Moore Z, Zeng D. Association of primary and 17 of 30 Curriculum Vitae Danyu Lin, Ph.D. January 15, 2024 booster vaccination and prior infection with SARS-CoV-2 infection and severe COVID-19 outcomes. JAMA 2022c; 328: 1415–1426.

Reviewer 2 Report

Comments and Suggestions for Authors

It is good to see the topic of immune imprinting included in this manuscript.  No response to this point is needed.

Lines 193-195, 235-242, 262-264, 313-315, etc. - These is are important observations; other similar important observations are included in this manuscript.  No response to this point is needed.

Line 386-387 - It is not clear that evidence or vaccination strategies have been presented for how to "maintain" robust immune responses.  My concern is on how to "maintain".

Author Response

Comment: Line 386-387 - It is not clear that evidence or vaccination strategies have been presented for how to "maintain" robust immune responses.  My concern is on how to "maintain".

Response:

Thanks for the constructive comment. In the conclusion section, we stated:

“While additional vaccine doses, particularly the bivalent Comirnaty vaccine, and hybrid immunity offer increased protection, the potential immune imprinting effect observed with the Comirnaty vaccine warrants further investigation to optimize vaccination strategies and maintain robust immune responses against emerging SARS-CoV-2 variants.”

In this study, we focused on the existence of immune imprinting effect based on the observed data. However, we do not have a clear solution for maintaining robust immune responses. 

The question on “How” to maintain robust immune responses requires further investigation as stated in the conclusion.  This is not a straightforward issue, as we need to define a statistical measure for the protection level to be maintained. Additionally, it is more realistic to consider the waning protective efficacy of the vaccines in order to develop a vaccination schedule that can sustain the protective efficacy at a certain level. This is indeed the question we are addressing in an ongoing project.